# Composition of Pinot Noir Wine from Grapevine Red Blotch Disease-Infected Vines Managed with Exogenous Abscisic Acid Applications

**DOI:** 10.3390/molecules27144520

**Published:** 2022-07-15

**Authors:** Ling Huang, Armando Alcazar Magana, Patricia A. Skinkis, James Osborne, Yanping L. Qian, Michael C. Qian

**Affiliations:** 1Department of Food Science and Technology, Oregon State University, Corvallis, OR 97331, USA; ling.huang@oregonstate.edu (L.H.); armandoalkazar@gmail.com (A.A.M.); james.osborne@oregonstate.edu (J.O.); yan.ping.qian@oregonstate.edu (Y.L.Q.); 2Department of Horticulture, Oregon State University, Corvallis, OR 97331, USA; patricia.skinkis@oregonstate.edu; 3Oregon Wine Research Institute, Oregon State University, Corvallis, OR 97331, USA

**Keywords:** grapevine red blotch disease, exogenous abscisic acid application, vineyard management, wine composition, wine volatile compounds, untargeted LC-HRMS/MS analysis

## Abstract

Grapevine red blotch disease (GRBD) has negative effects on grape development and impacts berry ripening. Abscisic acid (ABA) is a plant growth regulator involved in the initiation of berry ripening. Exogenous abscisic acid application was compared to an unsprayed control on GRBD-positive Pinot noir vines during two vintages, and the total monomeric anthocyanin, total phenolics, phenolic composition, and volatile profile were measured in wines. In addition, untargeted metabolites were profiled using high-resolution LC-MS/MS. Results showed that the wine composition varied by vintage year and was not consistent with ABA application. Wines from the ABA treatment had a lower total anthocyanin and total phenolic content in one year. The untargeted high-resolution LC-MS/MS analysis showed a higher abundance of phenolic compounds in ABA wines in 2019, but lower in 2018. The wine volatile compounds of ABA treatments varied by vintage. There were higher levels of free β-damascenone, β-ionone, nerol, and several fermentation-derived esters, acids, and alcohols in ABA wines, but these were not observed in 2019. Lower 3-isobutyl-2-methoxypyrazine (IBMP) was also observed in wines with ABA treatment in 2019. The results demonstrated that ABA application to the fruit zones did not consistently mitigate the adverse impacts of GRBD on Pinot noir wines.

## 1. Introduction

Grapevine red blotch disease (GRBD), an emerging grapevine disease caused by grapevine red blotch virus (GRBV), has aroused increasing attention in the wine industry worldwide. The virus has been found throughout vineyards in many countries, such as Argentina, Canada, India, Korea, Mexico, Switzerland, and the United States of America [1,2,3,4,5,6,7]. Both red and white *Vitis vinifera* grape species can be affected by GRBV [1]. The main symptoms of GRBD include red coloring of leaf blades and leaf veins in red-fruited cultivars, while chlorotic or necrotic leaf margins have been observed in white-fruited cultivars [8]. The foliar symptoms start at véraison in the basal leaves and are expressed in more apical leaves as the season progresses [8]. GRBD has a physiological effect on grapevines by reducing the leaf gas exchange, chlorophyll concentration, and vegetative growth in GRBV-positive grapevines [3]. Reduced leaf photosynthesis was also reported in GRBV-positive grapevines [9].

Secondary berry metabolites can also be influenced by GRBD. It has been reported that grapes from symptomatic red blotch disease grapevines were lower in total soluble solids, flavan-3-ol, and total phenolic content and higher in the flavonol content when compared to grapes from healthy, GRBV-negative grapevines [10]. Enzyme activities involved in core phenylpropanoid, stilbene, flavonoid, and anthocyanin biosynthetic pathways are reduced in GRBV-positive berries at ripening stages [11]. Except for hydroxycinnamic acid, all phenolic compounds present lower concentrations in skins and pulp tissues of berries from GRBV-positive vines [12]. A water deficit is typically an effective way to increase anthocyanins in grapes during ripening; however, a regulated deficit irrigation does not improve fruit quality in GRBV-positive grapevines [13,14]. In general, GRBV-positive vines have decreased sugar, oligosaccharides, and polyphenols in the berries. On the other hand, GRBD causes higher amino acids and organic acids in the berries. The effects of GRBD on grape composition differ between the cultivar, vineyard, and vintage [15].

GRBD can also affect the wine volatile composition and change wine quality. Girardello et al. [10] found wines produced from GRBV-positive vines had lower levels of α-terpinene, p-cymene, linalool, and limonene than wines created with grapes from healthy, GRBV-negative vines. Since linalool and limonene concentrations accumulate in grapes and reach the maximum levels at later stages of ripening [16], it is possible that the lower levels of linalool and limonene in wines were caused by the delayed accumulation of those compounds in the grapes due to GRBV.

Abscisic acid (ABA) is an important regulation phytohormone of berry maturation and ripening [17]. ABA might contribute to the pathogen defense system, as it decreases the presence of spoilage fungi on the surface of fruit [18]. ABA induces defense responses by upregulating the expression of some defense-related genes and phytoalexin synthesis in mature berries [19].

Anthocyanins contribute to grape berry color and are synthesized via the phenylpropanoid pathway. It has been reported that ABA can have a positive effect on grape phenolic compounds by increasing the total anthocyanin and total phenolic content in grape berry skin at harvest [20]. ABA may increase low-molecular-weight polyphenols and anthocyanins in final wines [21]. It has been demonstrated that exogenous ABA can modulate transcription factors involved in the phenylpropanoid pathway differently to increase the amount of petunidin- and malvidin- type anthocyanins, which improve the skin color of grapes [22].

Phenolic compounds are important for the grape and wine fruit color, astringency, and bitterness [23]. Phenolic compounds in grapes are generally synthesized in the berry and accumulated in the berry skin and seeds. The major phenolics in grapes are tannins, anthocyanins, hydroxycinnamates, flavan-3-ols, flavonols, and stilbenes/stilbene derivatives [24]. Phenolic compounds are synthesized from the amino acid phenylalanine through the phenylpropanoid pathway [25]. Stilbenes, like resveratrol, are produced in grapevine tissues as phytoalexins to mitigate biotic and abiotic stresses and serve to protect the plant from pathogens [26,27].

Wine volatile compounds are important for the wine aroma and quality. Volatile compounds can be sourced from grapes, fermentation, and the aging process [28]. Grape-derived compounds include pyrazines, terpenes, and norisoprenoids. Methoxypyrazines are nitrogenated heterocyclic products of the amino acid metabolism, which originate in the grape berry and are associated with green pepper and herbaceous aromas [29]. Terpenes contribute fruity and floral aromas to wines and are distinguishable between grape varieties [29]. C_13_-norisoprenoids are formed through carotenoid degradation and are present in grape berries as both free-form and nonvolatile glycosides, which can be transformed to the free-form through enzyme or acid hydrolysis [30,31]. Many norisoprenoids have an extremely low sensory threshold; thus, small changes in concentration can have a large impact on wine aroma. Esters and higher alcohols are predominant fermentation metabolites and are commonly present in high concentrations in wine [28].

In this study, we investigated the impact of exogenous ABA application to vines in situ compared to untreated GRBV-positive vines over the course of a two-year study. Anthocyanins, phenolics, and volatile aroma were evaluated in wines created from GRBV-infected grapes with or without ABA applied to the cluster zones. The goal was to determine whether exogenous ABA application could improve berry maturation in GRBV-positive host plants by interfering with normal hormone and metabolite transport, and improve the sugar, flavonoid, and anthocyanin content in the berry.

## 2. Results

### 2.1. Effects of ABA Vine Treatment on Anthocyanin and Phenolics of Grapevine Red Blotch Disease-Associated Wines

The wine monomeric anthocyanin and total phenolic contents across two seasons varied (Figure 1). Both levels of the monomeric anthocyanin and total phenolic content were lower in 2018 wines associated with ABA treatment compared to the CON wines, while no statistical difference was observed in 2019. An HPLC analysis of phenolic compounds in wine (Table 1) showed significant decreases in caffeoyltartaric acid and malvidin-3-monoglucoside for ABA treatment in 2018, and a significant decrease in trans-coutaric acid for the ABA treatment in 2019. All other phenolic compounds measured with HPLC did not change between ABA and CON.

### 2.2. Volatile Compounds

Wine volatile compounds were affected by cluster zone ABA application in this study, but the impact varied between the two vintages (Table 2). In 2018, ABA wines had higher levels of (*E*)-3-hexen-1-ol, 1-hexanol, 2-heptanol, benzyl alcohol, β-damascenone, β-ionone, ethyl decanoate, ethyl dodecanoate, and nerol than CON. The levels of isoamyl alcohol, 2-phenylethanol, linalool, bound-form 1,1,6-trimethyl-1,2-dihydronaphthalene (TDN), and vitispirane were lower in wines from ABA-treated fruit compared to CON. The ABA treatment caused higher total concentrations of esters, C_13_-norisoprenoids, and terpenes in 2018 GRBD wines (Table 2). In 2019, higher concentrations of propanol, ethyl decanoate, and ethyl undecanoate and lower concentrations of 3-methylbutanoic acid, hexanoic acid, 2-phenylethanol, ethyl 2-methylbutanoate, ethyl 2-methylpropanoate, ethyl 3-methylbutanoate, ethyl octanoate, ethyl propionate, isoamyl acetate, isobutyl acetate, phenethyl acetate, and 3-isobutyl-2-methoxypyrazine (IBMP) were observed in wines with the ABA treatment compared to CON.

The grape harvest year (vintage) played an important role in the GRBV-positive Pinot noir wine composition under ABA and CON treatments. Wines from each vintage were separated into two sides of the *y*-axis based on the F1 component, regardless of ABA or CON treatments (Figure 2). The two components explained 75.19% variance of the data with a 95% confidence interval.

### 2.3. Untargeted LC-MS/MS Fingerprinting

In total, more than one hundred compounds were tentatively identified with the untargeted LC-HRMS/MS analysis. These metabolites were classified as phenols (43%), amino acids and their derivatives (5 and 12%, respectively), organic acids (4%), lipids (4%), carbohydrates (3%), and several other metabolite categories (Figure 3). Phenolic compounds of interest concerning GRBD were further classified into flavonols (13%), flavan-3-ols (7%), anthocyanins (6%), phenolic acids (5%), stilbenes (4%), anthocyanidins (2%), flavones (1%), and other categories. A detailed list of tentative identifications from the untargeted metabolomic study is included in the Appendix A.

A principal component analysis (PCA) of LC-HRMS/MS data showed a greater separation between wine vintage (2018 and 2019) compared to treatment (ABA and CON), with no apparent separation of treatment groups within each year (Figure 4). This observation suggested no overall effect of ABA application on nonvolatiles in wines produced from GRBV-positive vines. Several compounds were found to differ between treatment groups, eight compounds in 2018 wines and eleven compounds in 2019 wines (Figure 5); however, no compounds were common across both years. Of the 19 compounds found to be different across treatment groups, six phenols were identified, including hyperoside, quercetin-3-rhamnose, procyanidin B2, trans-piceid, resveratrol, and isoquercetin. In 2018, total LC-HRMS/MS abundances of phenolic compounds were lower in the ABA treatment group. In contradiction, phenolic compounds increased with ABA treatment compared to the unsprayed control in 2019. The untargeted LC-HRMS/MS analysis showed the vintage year had a greater impact in the relative abundance of compounds on GRBV-positive wines than ABA treatments.

## 3. Discussion

Although ABA was reported to increase the anthocyanin concentration in grapes by adjusting the anthocyanin biosynthesis pathway [16], this was not observed in wine from the current study using GRBV-positive vines. ABA impacts may depend on the grape variety, vineyard, vintage weather conditions, or timing of application. Regardless, the current results indicated that ABA application at véraison did not mitigate the negative effect of GRBV for anthocyanin in 2018 Pinot noir wine (Figure 1). Both levels of the monomeric anthocyanin and total phenolic content were lower in 2018 wines associated with ABA treatment than CON. The monomeric anthocyanin and total phenolic content were maintained with the application of ABA to GRBV-positive vines and no statistical difference was observed in either phenolic measurement in 2019. Inconsistent results across the two-year study did not allow for confident conclusions on the overall effect of ABA on the phenolic content in GRBD. The varying levels of anthocyanin in ABA treatment may result from the complex interaction between ABA and GRBV, of which the mechanism remains unclear.

The volatile compounds which were statistically different between ABA treatments did not overlap between vintages, except for 2-phenylethanol and ethyl decanoate. Lower 2-phenylethanol and higher ethyl decanoate levels were found in ABA than CON in both years. As a higher alcohol, 2-phenylethanol can be formed through the amino acid metabolism majorly or glycosylated precursors during the vinification process [32]. The production of higher alcohols was reported to be negatively related to the yeast-assimilable nitrogen (YAN) level [33,34]. The grape basic chemical composition analysis showed higher YAN levels in GRBV-positive grapes under ABA treatment than in the unsprayed control [35]. In our study, the lower 2-phenylethanol level in the ABA treatment was in agreement with the high YAN in the berries.

Typically, in Pinot noir grapes, C6-alcohols accumulate before véraison, but their levels decrease during berry ripening [36]. Moreover, ABA was reported to be able to increase the C6 compound accumulation by inducing the gene expression of alcohol dehydrogenase, which converts aldehydes to alcohols [37]. This effect of ABA was observed in this study, where an increase in two C6 alcohols ((E)-3-hexen-1-ol and 1-hexanol) was found in the 2018 ABA treatment. In this study, the results in 2018 showed that the ABA treatment increased the total concentrations of free-form C_13_-norisoprenoids, esters, and terpenes in GRBD wines (Table 2). Statistical differences in norisoprenoid compounds were only found in 2018. The ABA treatment caused a decrease in bound-form norisoprenoids (vitispirane and TDN) and an increase in free-form norisoprenoids (β-damascenone and β-ionone). He et al. found similar effects of exogenous ABA on free-form norisoprenoids over two vintages of Cabernet Sauvignon [38].

The ABA treatment had a large effect on volatile esters in 2019 GRBD wines, where a significant decrease was observed in eight esters. GRBD was reported to reduce the concentration of esters and terpenes in final wines [6]. In this study, ABA application did not obviously mitigate the effects of GRBV on esters and terpenes. The decrease in IBMP in 2019 indicated that ABA may have an impact on grape ripening. The IBMP content found in Pinot noir grape decreased with increasing grape maturity, with the greatest decline in IBMP occurring before véraison [39].

The analysis of phenolic compounds with HPLC (Table 1) showed no advantage to ABA application with regard to increasing the phenolic content of GRBD-affected wines. The effect of ABA on increasing the concentrations of phenolics diminished by GRBD was not observed. In fact, the opposite effect was observed with three compounds (caffeoyltartaric acid, malvidin-3-o-glucoside, and trans-coutaric acid) decreasing in concentration across the two-year study. Phenolic compounds were the major category identified with positive-ionization LC-HRMS/MS (Figure 3). Increases in trans-resveratrol and piceid were reported in ABA-treated Malbec cultivars [21], which is consistent with the untargeted LC-MS/MS results in the 2019 wines (Figure 5) from our study. The untargeted LC-MS/MS analysis of wines in this study did not show the consistent effects of exogenous ABA on GRBV-positive Pinot noir.

The Pinot noir wine composition varied more by vintage than ABA treatment. Weather conditions can impact vine growth, plant water stress, and the mineral nutrient status, thereby impacting fruit and wine color and aroma profiles. Dry climatic conditions often increase anthocyanins, as well as the glycoconjugate forms of aroma compounds in grapes [40]. Year 2018 was dry (246 mm for the season) and warm (20.9 °C mean daily temp from bloom to véraison), while 2019 was cooler (19.6 °C mean daily temp from bloom to véraison) and wetter (328 mm rainfall for the season) with a lower average air temperature and greater precipitation (Aurora, Oregon AgriMet United States Bureau of Reclamation, https://www.usbr.gov (accessed on 8 April 2022)). The grapevines in 2018 showed more GRBD symptoms and had less overall vine growth compared to 2019, where vines had little to no visual symptoms of the virus and higher amounts of vine growth. The impact of ABA on wine composition during the more abiotic stress year (2018) may suggest a benefit only under conditions when the vines are under abiotic stress and symptomatic.

## 4. Materials and Methods

### 4.1. Chemicals

Ethanol (HPLC grade), methanol (HPLC grade), formic acid (HPLC grade), sodium chloride, and citric acid were purchased from Fisher Scientific (Santa Clara, CA, USA). All reference standard compounds used for major phenolic compound and volatile compound analyses were purchased from Sigma Aldrich (St. Louis, MO, USA) with at least 97% purity. Stable isotope standards were obtained from Cambridge Isotope Laboratories, Inc. (Tewksbury, MA, USA). Milli-Q water was used throughout the experiment.

### 4.2. Field Experiment

A two-year trial (2018 and 2019) was conducted in a commercial vineyard located north of Lafayette, Oregon. The vineyard was planted in 2000 with *Vitis vinifera* L. ‘Pinot noir’ clone 777 grafted on 3309C rootstock with vine spacing of 1.5 m rows and 2.4 m between rows with N–S orientation on a south-facing slope at 100 m above sea level on Goodin and Witham silty clay loam soils. Vines were cane pruned and trained to a bilateral Guyot system with vertical shoot positioning. Each season, the vineyard was managed using standard disease and canopy management practices for the region and was not irrigated. Grapevines within a 1.1-hectare (2.73 acres) block were tested and identified as being positive for grapevine red blotch virus (GRBV). Individual vines along grid sampling (*n* = 96) per plot were sampled and tested for GRBV in summer 2018 using leaf tissues, but were tested again to confirm virus status using dormant cane tissue following the crop year. All samples were analyzed at the USDA-ARS Horticultural Crops Research Unit Corvallis, OR using polymerase chain reaction (PCR) procedures [7].

Two treatments (ABA application and unsprayed control) were applied to whole vineyard rows. Each plot consisted of six rows, with five field replicates in a randomized complete-block design. The ABA treatment was sprayed with 300 mg/L Protone SG (ProTone SG, Valent Biosciences Corp, Libertyville, IL, USA) and 0.01% surfactant (Kinetic, Helena Agri-Enterprises, LLC, Collierville, TN, USA) using a canopy airblast sprayer with nozzles directed to the cluster zones only. The spray was applied until dripping at the onset of véraison (BBCH 80) and again two weeks later. Applications occurred on 11 August and 25 August in the 2018 season and on 9 August and 23 August in the 2019 season.

### 4.3. Winemaking

Grapes were collected 1 to 2 days prior to commercial harvest each year on 1 October 2018 and 2 October 2019, respectively. A total of 60 kg of fruit was harvested from each of the five field replication plots for each treatment. Wines were created in Oregon State University research winery. Harvested grapes were stored at 4 °C overnight in a cold room. The following day, the field replicates for each treatment were combined and destemmed. After destemming, grapes were mixed and weighed out to the target weight of 25 kg per plastic fermenter in six replicates. Samples from each fermenter were assessed for basic grape composition as described [35]. No significant differences between treatments for total soluble solids or pH were measured in 2018 and 2019, with only a small difference in titratable acidity (TA) being noted in 2019 [35]. Total soluble solids in 2018 ranged from 24.2 to 23.8 °Brix and in 2019 ranged from 21.2 to 21.4 °Brix. Measured pH ranged from 3.21 to 3.24 in 2018 and 3.07 to 3.05 in 2019. All fermenters received an addition of 50 mg/L SO_2_ (as potassium metabisulfite) and the yeast nutrient Fermaid K^®^ (Lallemand, Montreal, QC, Canada) was added at a rate of 0.25 g/L. Ferments were then inoculated with *Saccharomyces cerevisiae* RC212 (Lallemand, Montreal, QC, Canada) at a rate of 0.25 g/L following rehydration according to the manufacturer.

Fermentations were conducted in a heated room set at 27 °C. Manual punch downs were performed twice daily prior to total soluble solids and temperature measurements (data not shown) using an Anton-Paar DMA 35 N Density Meter (Graz, Austria). At the completion of fermentation (14 days), treatments were pressed at 0.1 MPa for 5 min and the wine was settled overnight at 4 °C. Wine was racked into sanitized 3-gallon carboys and warmed to 21 °C prior to inoculation with *Oenococcus oeni* VP41 (Lallemand, Montreal, Canada) at approx. 10^6^ CFU/mL to induce malolactic fermentation (MLF). At the completion of MLF (24 days, malic acid < 100 mg/L as measured by enzymatic assay (Vintessential, Victoria, Australia)), 50 mg/L SO_2_ (as potassium metabisulfite) was added. Wines were settled at 4 °C before being racked and topped to minimize headspace. Samples of the individual replicates were taken and assessed for basic wine chemistry analysis as described [35]. No differences in wine pH, TA, or ethanol were noted between treatment wines in 2018 or 2019 [35]. An additional 100 mL wine sample from each fermentation replicate under each treatment was taken in 50 mL centrifuge tubes and stored at −20 °C before analysis.

### 4.4. Wine Monomeric Anthocyanin

Total monomeric anthocyanin was determined with a UV-1800 spectrophotometer (Shimadzu, Kyoto, Japan) according to the absorbance change of anthocyanin at pH 1.0 and pH 4.5 [41,42]. Wine samples were diluted ten-fold with milli-Q water, then 0.5 mL of the diluted sample was mixed with 2 mL 0.4 M sodium acetate buffer (pH 4.5) and 2 mL 0.025 M potassium chloride solution (pH 1.0, adjusted with HCl) separately. Mixtures were placed in cuvettes, and absorbance was measured at 520 nm and 700 nm. The concentration of monomeric anthocyanin was calculated based on the absorbance difference of pH 1.0 and pH 4.5. Samples were analyzed in triplicate and expressed as mg/L malvidin-3-glucoside equivalent.

### 4.5. Total Phenolic Content

Folin–Ciocalteu colorimetric method was used to measure wine total phenolic content [42]. Briefly, 100 µL wine sample and 500 µL Folin–Ciocalteu reagent (Sigma Aldrich, St. Louis, MO, USA) were diluted to 6 mL with milli-Q water in a glass tube. The solution was reacted for 5 min followed by the addition of 4 mL 7.5% sodium carbonate solution (*w*/*v*), and placed in darkness at room temperature for 2 h. Gallic acid was prepared at several levels (40, 80, 120, 160, 200 mg/L) as a standard curve. Absorbance was measured at 740 nm with UV-1800 spectrophotometer (Shimadzu, Kyoto, Japan). Samples were analyzed in triplicate and expressed as mg/L gallic acid equivalent.

### 4.6. Major Phenolic Composition

High-performance liquid chromatography (HPLC) was used to determine phenolic composition in wines [39]. Wine samples were filtered with 0.45 μm nylon membrane (PALL, Washington, NY, USA). Five microliters of the filtrate was injected into the HPLC system. The HPLC system was equipped with an Agilent 1100 series diode array detector (Palo Alto, CA, USA). Data acquisition and processing were completed using ChemStation software (Agilent, v.10.02). Separation was carried out on a Kinetex XB-C18 column (100 Å, 2.6 μm, 150 × 4.6 mm, Phenomenex, Torrance, CA, USA) under wavelengths of 280 nm, 360 nm, and 520 nm. Solvent A (0.5% formic acid in milli-Q water) and solvent B (100% methanol) were used as mobile phases under a flow rate of 0.8 mL/min. Solvent gradient: 0–5 min (3–16% B); 5–15 min (16–35% B); 15–20 min (35–55% B); 20–25 min (55–100% B); 25–27 min (100% B); 27–29 min (100–3% B); 29–35 min (3% B). The phenolic compounds (caffeic acid, caffeoyltartaric acid, catechin, epicatechin, fertaric acid, gallic acid, malvidin-3-o-glucoside, p-coumaric acid, resveratrol, trans-coutaric acid, and vanillic acid) were identified through comparison with UV spectra and retention times of analytical standards. Quantitation was conducted through external calibration curves. Caffeic acid, caffeoyltartaric acid, catechin, epicatechin, fertaric acid, gallic acid, p-coumaric acid, resveratrol, trans-coutaric acid, and vanillic acid were quantified under 280 nm and malvidin-3-glucoside was quantified under 520 nm. Wavelength 360 nm was used as an additional reference for the identification of phenolic acids.

### 4.7. Wine Volatile Analysis

Wine volatile compounds were analyzed with headspace gas chromatography-flame ionization detector (HS-GC-FID) and headspace solid-phase microextraction gas chromatography mass spectrometry (HS-SPME-GC-MS) [43]. Each sample was analyzed in triplicate.

Highly volatile compounds, including acetaldehyde, ethyl acetate, propanol, isobutanol, and isoamyl alcohol were analyzed through HS-GC-FID. In brief, 0.5 mL wine was diluted two-fold with Milli-Q water and 20 μL methyl propionate (2.5 mg/L) was added as an internal standard. Extraction and injection were conducted in a Gerstel MPS (multipurpose sampler, Linthicum, MD, USA) autosampler equipped with a 2.5 mL headspace syringe. Before injection, samples were incubated in a thermostatic bath at 50 °C for 15 min. The injection volume was 0.5 mL with a 1:10 split ratio. Analysis was performed on Agilent 6890 N GC coupled with a flame ionization detector (Agilent Technologies Inc., Santa Clara, CA, USA). Compounds were separated on a ZB-WAX column (30 m × 0.25 mm i.d., 0.5 μm film thickness, Phenomenex Inc., Torrance, CA, USA). The column flow rate was 1.5 mL/min with helium as the carrier gas. The initial oven temperature was 35 °C held for 4 min, then increased by 10 °C/min to 150 °C and held at 150 °C for 5 min. Injection port temperatures was 250 °C.

Major volatile compounds were analyzed with HS-SPME-GC-MS. Two-milliliter wine sample was diluted 5-fold with 0.2 M citrate buffer (pH 3.5) saturated with sodium chloride. Isotope-labeled internal standard solution and a magnetic stir bar were added to the vial. Sample vials were equilibrated at 50 °C in a thermostatic bath for 10 min and then the headspace was extracted with a 2 cm divinylbenzene/carboxen/polydimethylsiloxane (DVB/CAR/PDMS, 50/30 μm film thickness, Supelco, Bellefonte, PA, USA) SPME fiber for 50 min at the same temperature under stirring (500 rpm). Desorption was performed by introducing the SPME fiber into the GC injection port for 5 min with splitless mode. Analysis was performed on Agilent 6890 N GC coupled with a 5973 N mass selective detector (Agilent Technologies Inc., Santa Clara, CA, USA). Compounds were separated on a ZB-WAX column (30 m × 0.25 mm i.d., 0.5 μm film thickness, Phenomenex Inc., Torrance, CA, USA). The column flow rate was 1.5 mL/min with helium as the carrier gas. Initial oven temperature was 40 °C and held for a 4 min, then increased to 230 °C at a rate of 4 °C/min, with a 10 min hold at the final temperature. Methoxypyrazines analysis was conducted on a HP-5 column (30 m × 0.25 mm i.d., 0.5 μm film thickness) under SIM. The following oven program was used for methoxypyrazines: initial oven temperature was 40 °C and held for a 2 min, increased to 120 °C at a rate of 3 °C/min, then increased to 230 °C at a rate of 6 °C/min with a 10 min hold at the final temperature. Monitored selected ions: 2-isopropyl-3-methoxypyrazine (IPMP), *m/z* 137; 2-sec-butyl-3-methoxypyrazine (SBMP), *m/z* 138; 3-isobutyl-2-methoxypyrazine (IBMP), *m/z* 124; IBMP-d_3_, *m/z* 127. Injection port, MS transfer line, and ion source temperatures were 250, 280, and 230 °C, respectively. Electron ionization mass spectrometric data from *m*/*z* 40 to 250 were collected, with an ionization voltage of 70 eV.

Compound identification was performed by comparison to retention times and matching mass spectra of analytical standards under identical conditions. Compounds were quantified with internal standard calibration curves which were built with gradient dilution of known amounts of the standards. Isotope-labeled internal standards used for major volatile compounds analysis included ethyl butyrate-4,4,4-d_3_, ethyl 2-methyl butyrate-d_9_, ethyl 3-methyl butyrate-d_9_, ethyl hexanoate-d_11_, nonanal-d_18_, ethyl octanoate-d_15_, linalool-d_3_, α-terpeniol-d_3_, ethyl decanoate-d_19_, hexanoic acid-d_11_, 2-phenyl-d_5_-ethan-1,1,2,2-d_4_-ol, octanoic acid-d_15_, and decanoic acid-d_19_. The isotope internal standard with similar structure or the nearest retention time was used if the corresponding isotope was not available. Identification and calculation were processed with ChemStation software (ver. E.02, Agilent Technologies Inc., Santa Clara, CA, USA).

### 4.8. Wine C_13_-Norisoprenoids (Bound-Form) Analysis

Wines were hydrolyzed in acidic condition and analyzed for bound-form C_13_-norisoprenoids with HS-SPME-GC-MS based on the previous method with minor modifications [42]. Two-milliliter wine sample was diluted 5-fold with 0.2 M citrate buffer (pH 2.5) saturated with sodium chloride in a 20 mL vial. The vial was placed in a shaking water bath (99 °C, 1 h) for acid hydrolysis. After hydrolysis, the sample was placed in an ice bath for 5 min before the addition of internal standard ((±)-theaspirane, 1 mg/L). Internal standard calibration curves were built with devolatilized Pinot noir wine. Devolatilized Pinot noir wine was generated by passing wine through a solid-phase extraction column with 2 g Lichrolut EN (Merck, Darmstadt, Germany). Standard mixtures of C_13_-norisoprenoids were added and diluted to different gradients. HS extraction conditions and GC method were the same as for the analysis of major volatile compounds.

### 4.9. Untargeted Analysis with Liquid Chromatography High-Resolution Tandem Mass Spectrometry (LC-HRMS/MS)

Wine samples were diluted ten times with aqueous methanol 50% *v/v* before injecting 3 µL into the system. Untargeted analysis of wine was performed according to a previously reported fingerprinting method with some modifications [44]. LC-HRMS/MS analysis was conducted using a UHPLC system (Shimadzu Nexera, Shimadzu Corporation, Kyoto, Japan) coupled to an AB SCIEX Triple TOF^®^ 5600 mass spectrometer (Framingham, MA, USA) operated in positive electrospray ionization (ESI) mode. Chromatographic separation was performed utilizing an Inertsil Phenyl-3 column (4.6 mm × 150 mm, 100 Å, five μm; GL Sciences, Torrance, CA, USA). Mobile phase A consisted of water containing 0.1% *v/v* formic acid, and mobile phase B, methanol containing 0.1% *v/v* formic acid. The flow rate was 0.4 mL/min. The entire run was 30 min in a multistep gradient as follows: 0–1 min at 5% B, followed by 5 to 30% B from 1 to 10 min, then 30 to 100% B from 10 to 20 min, held at 100% B from 20 to 25 min and then returned to 5% B from 25 to 30 min. For each year, a quality control (QC) sample was prepared by mixing an equal volume of each wine sample. In addition, QC was injected every two hours to monitor system stability.

Data-dependent acquisitions (DDAs) were conducted to obtain precursor and fragment ion information to aid in annotating compounds. The following settings were used: spray voltage was set to 4500 V, full scan with ion accumulation of 150 ms, followed by a dynamic MS/MS selection of the eight most intense ions with 100 ms accumulation; after two MS/MS acquisitions, the precursor (fragmented) ions were excluded for 30 s; collision energy 35 V with collision energy spread (CES) of 15 V ramped through each MS/MS scan using a range of *m*/*z* 100–1200. In addition, the instrument was equipped with an automatic calibrant delivery system performing calibration every two hours. Annotation of compounds was performed as previously reported with some modifications [44,45,46]. In brief, exact mass, isotopic pattern, and MS/MS spectra were compared with online libraries: MassBank of North America (MoNA), METLIN, and Human Metabolome Database (HMDB) (online versions, July 2021). For compound annotation, a confidence score equal to or higher than 50 was considered adequate for putative annotation. L2 annotations were performed according to Sumner et al. [47]. Because this score was more rigorous than previous reports using Progenesis QI™ with a score > 31.6 [48], in the case of compounds previously reported in wine, the score selected was equal to or higher than 35. A high confidence score was typically reached when MS/MS similarity was >50%, the isotopic similarity was above 80%, and the exact mass error was better than 10 ppm.

### 4.10. Statistical Analyses

Two-sample *t*-test was conducted to determine if there were statistical differences for levels of monomeric anthocyanin, total phenolic content, and volatile compounds in GRBD Pinot noir wines with or without abscisic acid application. In addition, principal component analysis (PCA) was performed on the wine composition to visualize the wine samples across two seasons. Analyses were conducted with XLSTAT (Addinsoft, NY, USA).

## 5. Conclusions

ABA applied to the cluster zones reduced the total anthocyanin and total phenolics content in final wines associated with one year (2018). The HPLC analysis of phenolics showed that ABA application did not overcome the effects of GRBD. As anticipated, vintage had a greater effect on the volatile profile of wines. The effects of ABA on GRBD wine aroma compounds were year-dependent. The ABA treatment increased the total concentrations of esters, C_13_-norisoprenoids, and terpenes in the 2018 wines only. Lower levels of 2-phenylethanol were found consistently in ABA treatments in both years. Inconsistent results observed in the two-year study for phenolics and aroma profiles suggest ABA had no outstanding effect on GRBD. The untargeted analysis showed that the vintage year had a greater impact on the GRBD Pinot noir wine phenolic compounds than ABA application. Further work is still required to better understand the impact of exogenous ABA application on various cultivars, and its potential as a GRBD remediation treatment in infected vineyards. The continuation of the study to include multiple vintages and vineyard locations would determine a more robust data set, and allow for better conclusions on the effects of ABA on GRBD.

## Figures and Tables

**Figure 1 molecules-27-04520-f001:**
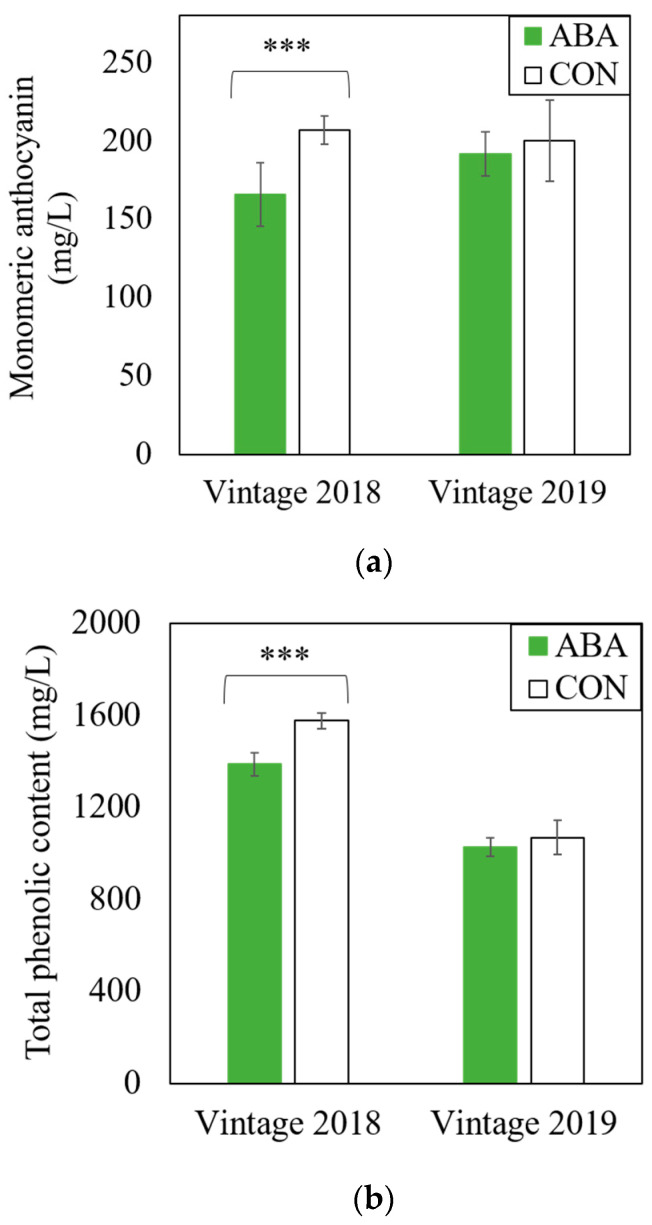
Effects of exogenous abscisic acid application (ABA) to the fruit zone and no abscisic acid control (CON) on levels of monomeric anthocyanin (**a**) and total phenolic content (**b**) in wines produced from GRBV-positive Pinot noir vines. Two-sample *t*-test was conducted for *p*-value calculation between ABA and CON in each year (***: *p* < 0.001).

**Figure 2 molecules-27-04520-f002:**
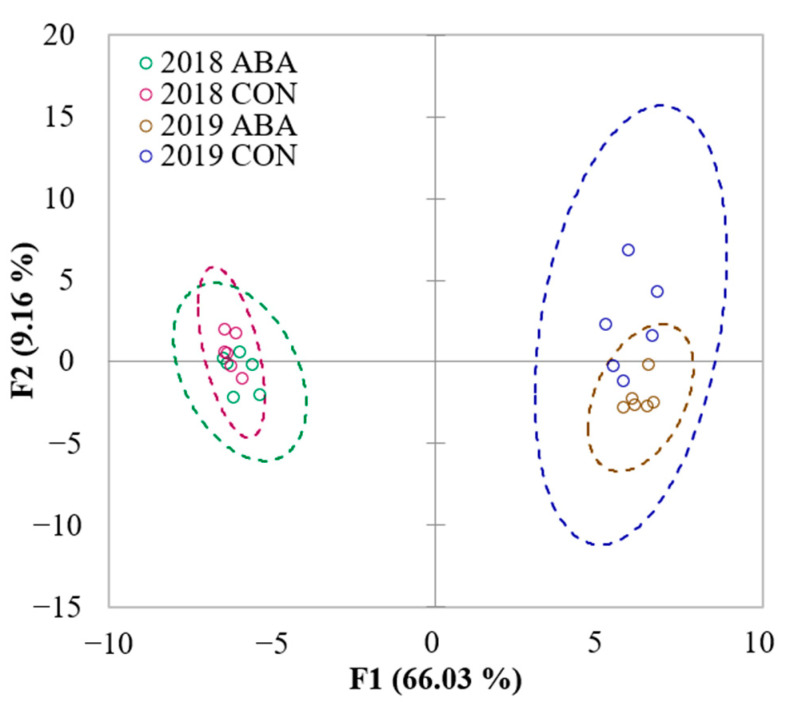
Score plot of principal component analysis on wines from GRBV-positive Pinot noir under exogenous abscisic acid (ABA) application and no abscisic acid control (CON) across two seasons (*n* = 6) in Oregon’s Willamette Valley. The plot was generated using the wine volatile profiles. Dash circles represent 95% confidence interval.

**Figure 3 molecules-27-04520-f003:**
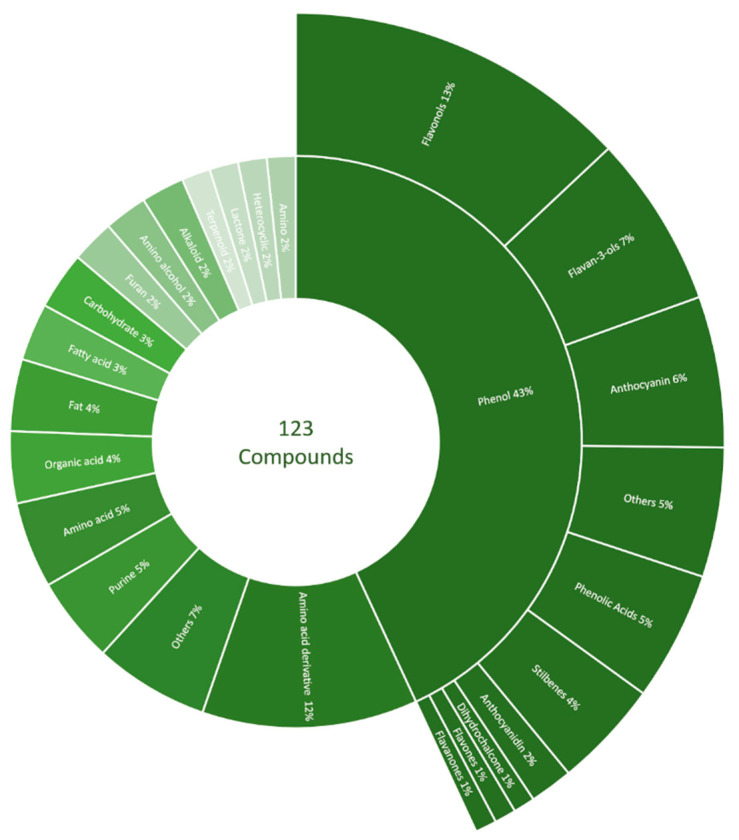
Percentage of putative annotation detected and identified by MassBank of North America (MoNA), METLIN, and Human Metabolome Database (HMDB) using LC-HRMS/MS. One hundred and twenty-three compounds were grouped into phenol, amino acid derivatives, others, purine, amino acid, organic acid, fat, fatty acid, carbohydrate, furan, amino alcohol, alkaloid, terpenoid, lactone, heterocyclic, and amino. Phenols were further divided into flavonols, flavan-3-ols, anthocyanin, others, phenolic acids, stilbenes, anthocyanidin, dihydrochalcone, flavones, and flavanones.

**Figure 4 molecules-27-04520-f004:**
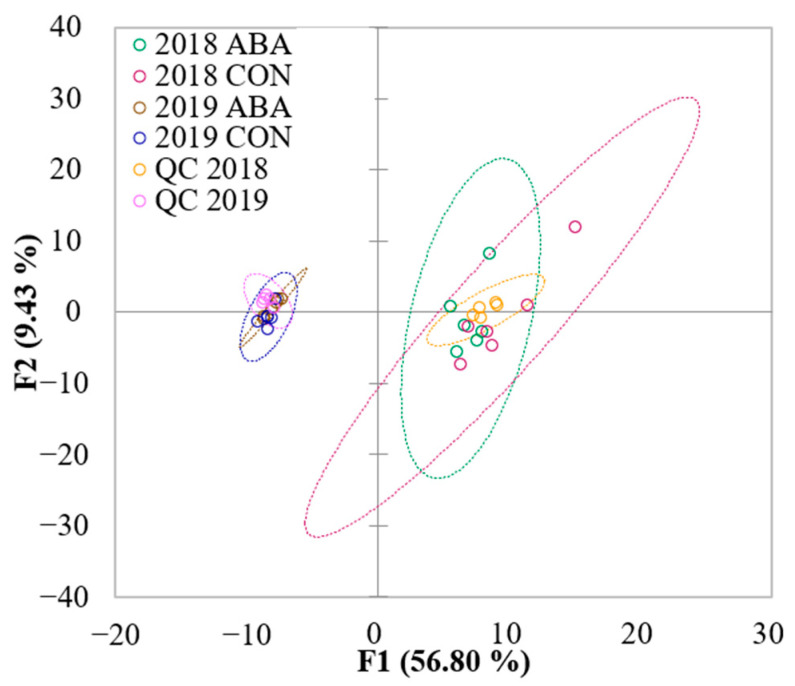
Score plot of principal component analysis with untargeted analysis data obtained with LC-HRMS/MS on wines from GRBV-positive Pinot noir treated with exogenous abscisic acid application (ABA) and no abscisic acid control (CON) across two vintages (QC: quality control samples; *n* = 6) in Oregon’s Willamette Valley. Dash circles represent 95% confidence interval.

**Figure 5 molecules-27-04520-f005:**
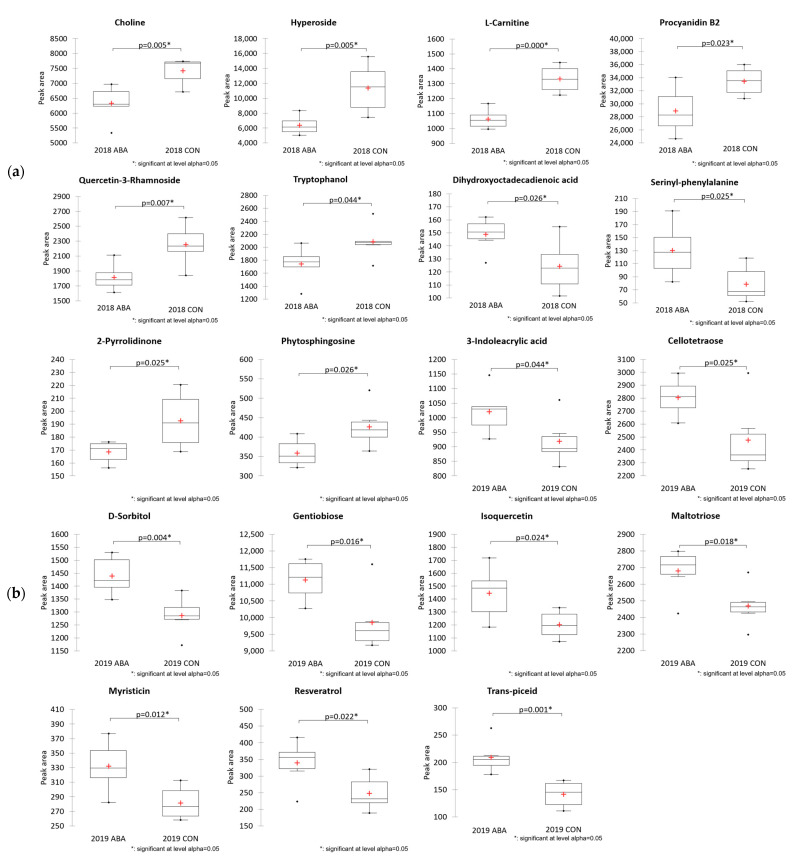
Boxplots of the tentatively annotated metabolites that differed statistically (*p* < 0.05) in wines from GRBV-positive Pinot noir treated with exogenous abscisic acid application (ABA) and no abscisic acid control (CON) for two years ((**a**): 2018; (**b**): 2019). *Y*-axis shows the relative abundance (peak area) of the compound. The box represents the 25% and 75% percentiles. The middle line of the box indicates the median and the red cross expresses the mean of the data. The short lines above and below the box represent the maximum and minimum points. Black dots on the two ends of the box represent single data. Two-sample *t*-tests were conducted for *p*-value calculation.

**Table 1 molecules-27-04520-t001:** Major phenolic compounds measured in wines from GRBV-positive Pinot noir vines under exogenous abscisic acid application (ABA) to the fruit zones and no abscisic acid control (CON) during two seasons.

Compound ^1^	Vintage 2018			Vintage 2019		
	ABA	CON	*p*-Value ^2^	ABA	CON	*p*-Value ^2^
caffeic acid	3.8 ± 0.3	4.2 ± 0.4	0.214	4.1 ± 1.4	5.0 ± 0.8	0.598
caffeoyltartaric acid	20.3 ± 0.7 ^a 3^	23.6 ± 0.5 ^b^	< 0.001	9.7 ± 5.4	13.1 ± 3.1	0.219
catechin	67.0 ± 6.0	71.8 ± 6.2	0.093	34.3 ± 2.6	35.3 ± 3.3	0.183
epicatechin	83.6 ± 7.5	81.0 ± 5.0	0.500	91.4 ± 15.0	83.5 ± 25.4	0.526
fertaric acid	7.2 ± 1.3	7.3 ± 1.3	0.901	1.7 ± 0.1	1.6 ± 0.1	0.322
gallic acid	9.4 ± 0.8	9.3 ± 0.5	0.741	6.5 ± 0.5	6.8 ± 0.4	0.275
malvidin-3-o-glucoside	124 ± 2 ^a^	141 ± 10 ^b^	0.002	263 ± 14	266 ± 21	0.732
p-coumaric acid	3.9 ± 0.4	3.8 ± 0.8	0.899	4.3 ± 1.1	4.9 ± 1.0	0.389
resveratrol	<1	<1		<1	<1	
trans-coutaric acid	8.3 ± 1.2	8.0 ± 0.3	0.517	9.6 ± 0.7 ^a^	10.8 ± 0.5 ^b^	0.009
vanillic acid	10.1 ± 1	11.4 ± 1.5	0.127	<1	<1	

^1^: Compound concentration is expressed as mean ± standard deviation in mg/L; ^2^: two-sample *t*-test was conducted for *p*-value calculation; ^3^: different letters between ABA and CON treatment under each year indicate the statistical difference (*p* < 0.05).

**Table 2 molecules-27-04520-t002:** Volatile compounds in wines produced from GRBV-positive Pinot noir vines under exogenous abscisic acid application (ABA) and no abscisic acid control (CON) during two vintages.

Compound ^1^	Vintage 2018			Vintage 2019		
	ABA	CON	*p*-Value ^2^	ABA	CON	*p*-Value ^2^
**acid**						
3-methylbutanoic acid ^3^	nd ^4^	nd		1.2 ± 0.1 ^a 5^	1.4 ± 0.1 ^b^	0.044
decanoic acid	269 ± 12	251 ± 18	0.068	141 ± 8	143 ± 8	0.680
hexanoic acid	395 ± 21	400 ± 8	0.580	1226 ± 41 ^a^	1352 ± 113 ^b^	0.028
octanoic acid	902 ± 35	890 ± 49	0.653	772 ± 28	837 ± 73	0.070
total	1566	1541		3339	3732	
**alcohol**						
(E)-2-hexen-1-ol	47.4 ± 12.8	58.5 ± 11.5	0.146	624 ± 109	650 ± 151	0.478
(E)-3-hexen-1-ol	135 ± 17 ^a^	111 ± 11 ^b^	0.013	326 ± 40	304 ± 58	0.752
(Z)-3-hexen-1-ol	31.7 ± 9.3	22.1 ± 7.5	0.080	52.4 ± 9.5	47.8 ± 10.0	0.496
1-hexanol ^3^	3.1 ± 0.3 ^a^	2.5 ± 0.2 ^b^	0.004	8.8 ± 1.2	10.6 ± 2.3	0.147
1-octen-3-ol	3.6 ± 0.4	3.2 ± 0.9	0.415	4.2 ± 0.4	4.3 ± 0.4	0.670
2-heptanol	6.8 ± 0.9 ^a^	4.7 ± 0.7 ^b^	0.001	11.7 ± 3.0	12.6 ± 1.5	0.534
benzyl alcohol	947 ± 34 ^a^	890 ± 41 ^b^	0.026	825 ± 43	780 ± 73	0.218
isoamyl alcohol ^3^	341 ± 24 ^a^	378 ± 11 ^b^	0.006	123 ± 12	128 ± 21	0.703
isobutyl alcohol ^3^	144 ± 20	159 ± 12	0.132	310 ± 24	316 ± 27	0.644
2-phenylethanol ^3^	39.5 ± 1.7 ^a^	41.9 ± 1.7 ^b^	0.036	27.5 ± 1.0 ^a^	35.0 ± 6.7 ^b^	0.022
propanol ^3^	23.0 ± 1.9	21.9 ± 2.8	0.472	37.0 ± 2.6 ^a^	32.5 ± 2.2 ^b^	0.009
total	551,772	604,390		508,143	523,899	
**aldehyde and ketone**						
6-methyl-5-hepten-2-one	nd	nd		2.2 ± 0.3	2.3 ± 0.8	0.688
acetaldehyde ^3^	11.5 ± 1.6	11.9 ± 0.5	0.561	nd	nd	
total	11,500	11,900		2.2	2.3	
**C_13_-norisoprenoid (free-form)**						
vitispirane ^6^	nd	nd		2.5 ± 0.1	2.8 ± 0.5	0.185
β-damascenone	5.7 ± 0.2 ^a^	5.4 ± 0.2 ^b^	0.024	5.2 ± 0.2	5.8 ± 0.9	0.135
β-ionone	0.6 ± 0.0 ^a^	0.5 ± 0.0 ^b^	0.015	3.0 ± 0.0	3.1 ± 0.1	0.351
total	6.3	5.9		10.7	11.7	
**C_13_-norisoprenoid (bound-form)**						
TDN ^7^	10.6 ± 0.8 ^a^	14.8 ± 2.3 ^b^	0.002	31.6 ± 3.2	31.5 ± 4.4	0.959
vitispirane ^6^	32.6 ± 4.0 ^a^	41.3 ± 4.2 ^b^	0.004	101 ± 7	105 ± 13	0.563
β-damascenone	16.2 ± 1.6	15.9 ± 2.2	0.764	61.8 ± 4.2	60.2 ± 8.1	0.666
β-ionone	0.6 ± 0.1	0.7 ± 0.1	0.111	2.5 ± 0.5	3.1 ± 0.7	0.134
total	60	73		197	200	
**ester**						
ethyl 2-methylbutanoate	5.0 ± 0.2	5.1 ± 0.4	0.500	3.9 ± 0.6 ^a^	5.1 ± 0.3 ^b^	0.001
ethyl 2-methylpropanoate	97.6 ± 10.1	100.4 ± 8.7	0.621	62.6 ± 5.4 ^a^	73.2 ± 8.2 ^b^	0.024
ethyl 3-methylbutanoate	6.4 ± 0.3	6.4 ± 0.3	0.755	5.7 ± 0.5 ^a^	6.6 ± 0.6 ^b^	0.027
ethyl acetate ^3^	57.4 ± 2.5	54.4 ± 3.8	0.135	32.1 ± 0.8	32.4 ± 1.3	0.618
ethyl butanoate	72.0 ± 4.5	69.9 ± 5.3	0.471	84.6 ± 3.2	92.7 ± 8.9	0.064
ethyl decanoate	27.1 ± 2.5 ^a^	24.0 ± 1.5 ^b^	0.027	35.6 ± 2.5 ^a^	29.5 ± 2.6 ^b^	0.002
ethyl dodecanoate	27.9 ± 6.6 ^a^	20.0 ± 3.8 ^b^	0.030	12.6 ± 1.1	13.0 ± 3.1	0.811
ethyl hexanoate	183 ± 9	165 ± 21	0.086	235 ± 15	219 ± 14	0.078
ethyl octanoate	55.3 ± 7.6	56.1 ± 7.2	0.847	72.9 ± 2.6 ^a^	81.3 ± 7.3 ^b^	0.028
ethyl phenylacetate	4.2 ± 0.7	5.0 ± 0.8	0.101	1.1 ± 0.1	1.2 ± 0.1	0.081
ethyl propionate	75.0 ± 8.5	76.5 ± 7.2	0.753	95.7 ± 4.9 ^a^	108.6 ± 9.9 ^b^	0.017
ethyl undecanoate	nd	nd		5.3 ± 0.1 ^a^	5.0 ± 0.3 ^b^	0.035
hexyl acetate	6.4 ± 1.0	6.0 ± 1.0	0.490	9.0 ± 0.4	8.6 ± 1.8	0.535
isoamyl acetate	386 ± 62	364 ± 27	0.430	391 ± 9 ^a^	481 ± 67 ^b^	0.009
isobutyl acetate	5.3 ± 0.2	5.2 ± 0.3	0.557	61.1 ± 2.4 ^a^	71.6 ± 9.9 ^b^	0.031
phenethyl acetate	25.7 ± 3.8	27.3 ± 2.5	0.404	20.1 ± 0.5 ^a^	21.8 ± 1.6 ^b^	0.039
total	58,377	55,331		33,196	33,618	
**lactone**						
γ-decalactone	1.6 ± 0.1	1.6 ± 0.1	0.181	nd	nd	
δ-undecalactone	2.7 ± 0.4	2.6 ± 0.2	0.539	nd	nd	
total	4.3	4.2		0	0	
**methoxypyrazine**						
IBMP ^8^	2.7 ± 0.7	2.6 ± 0.9	0.745	2.3 ± 0.3 ^a^	2.9 ± 0.3 ^b^	0.013
IPMP ^9^	2.4 ± 0.5	2.7 ± 0.2	0.207	2.0 ± 0.3	2.2 ± 0.6	0.488
SBMP ^10^	15.5 ± 2.2	15.9 ± 1.8	0.781	22.9 ± 1.5	20.9 ± 4.5	0.334
total ^11^	21	21		27	26	
**terpene**						
citronellol	11.6 ± 0.7	11.6 ± 0.7	0.940	25.7 ± 0.5	25.5 ± 0.7	0.532
geraniol	26.9 ± 0.6	26.4 ± 1.1	0.346	2.9 ± 0.4	3.6 ± 1.0	0.159
linalool	8.4 ± 0.3 ^a^	9.6 ± 0.8 ^b^	0.006	7.7 ± 0.2	9.3 ± 2.1	0.084
nerol	9.1 ± 2.9 ^a^	5.1 ± 2.1 ^b^	0.020	3.0 ± 0.1	3.8 ± 1.1	0.129
α-terpinol	3.8 ± 0.2	4.0 ± 0.6	0.524	4.5 ± 0.6	4.9 ± 1.1	0.399
total	60	57		44	47	

^1^: Compound concentration is expressed as mean ± standard deviation in µg/L; ^2^: two-sample *t*-test was conducted for *p*-value calculation; ^3^: concentration is expressed as mg/L; ^4^: not detected; ^5^: different letters between ABA and CON treatment under each year indicate the significant differences (*p* < 0.05); ^6^: concentration is expressed as β-damascenone equivalent; ^7^: 1,1,6-trimethyl-1,2-dihydronaphthalene; ^8^: 3-isobutyl-2-methoxypyrazine, ng/L; ^9^: 2-isopropyl-3-methoxypyrazine, ng/L; ^10^: 2-sec-butyl-3-methoxypyrazine, ng/L; ^11^: ng/L.

## Data Availability

The data presented in this study are available on request from the corresponding author.

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
