# Peer review of "Composition of Pinot Noir Wine from Grapevine Red Blotch Disease-Infected Vines Managed with Exogenous Abscisic Acid Applications"

_molecules, 2022, doi:10.3390/molecules27144520_

Round 1

Reviewer 1 Report

The reviewed work is not a typical chemical work, nevertheless it is very interesting and provides very important, practical conclusions. 

The research methodology does not raise my doubts. The assessment of errors and the limits of detection and quantification is appropriate. The discussion of the results is based on experimental data and reliable reasoning. From the analytical point of view (analytical methods, conditions of analysis, etc.), I rate the work highly. The authors performed a proper statistical analysis of the obtained results. They applied PCA analysis and obtained valuable conclusions. The presented conclusions are very cautious and the authors suggest the need for further research in this field. I believe that the article is worth publishing due to the fact that the subject (research object) is innovative, the results of numerous studies and analyzes are described, the authors have accumulated a number of reliable results that they have correctly interpreted. In my opinion, some letters in Tables 1 and 1S should be superscript. The work requires slight corrections in terms of typos.

Author Response

Thank you for your kind suggestions. We have made correction according to the comments on tables. Letters used for statistical differences were made superscript and other symbols used in table footnote were changed to numbers for clarity. Grammatical errors were corrected throughout the manuscript under track change mode.

Reviewer 2 Report

Excellent and great work. Of course, further work in this area is still required to better understand the application of exog. abscisic acid in vineyards. It's still had scientific challenges!

Author Response

Thank you for your kind comments. Our research group hope to continue working on this area.

Reviewer 3 Report

The manuscript presents the evaluation of wines produced from grapes treated with ABA and untreated (control). I believe the authors have taken some risks in having only two vintages in the study - the risk is high to have very different results and no way of saying which set is the most representative in the long run (which was the case here). The evident vintage effect observed for the various sets of analysis results is not only normal, but overrides almost any treatment effect. Again, with only two vintages, this split by vintage is to be expected. Moreover, in a lot of the literature, observed effects of vineyard treatments diminish with the amount of processing done to the initial sample (in this case, winemaking with both AF and MLF). Unfortunately, not even simple measurements on the grapes at harvest were presented in the manuscript.

In addition, the vineyard repeats were blended in the cellar before the batches were split up as winemaking repeats. This is again risky, since more extreme/obvious effects can be dampened. I would recommend the authors address in the manuscript/comment on the decision to blend the vineyard batches.

Oenological parameters have to be include in the manuscript - sugar, pH, TA for the grapes, alcohol, pH, TA for the wines. Even though the grape batches were blended, it is good practice to report these values. 

The authors have done various chemical analyses for the wines - the choice for the classes of compounds analysed should be better justified. (Some of this information appears in the Discussion section rather than in the Introduction) As it stands now, the various analyses performed could be read as simply a list of what was available in the lab. Even then, using a method that gives results for five phenolics can be seen as very basic compared to other methods available in the literature and to the extensive volatile compounds list.

The 'untargeted metabolomics' should be rephrased simply as 'untargeted LCMS analysis' (untargeted analysis followed by tentative identification). There is no metabolomics part included in the manuscript and indeed with the limited number of samples and treatments, that would be a stretch. Already the inclusion of PCA can be debated but at least in that case the observations x variables matrix is more balanced for the targeted analyses.

Author Response

We would like to express our great appreciation to your constructive comments on our paper. We have addressed all comments carefully and revised the manuscript accordingly.  detailed response was attached in the fiile.

Round 2

Reviewer 3 Report

I would like to commend the authors for the effort they put into reworking the manuscript.